# OpenReview forum: "Vision-Language Models Are Not Pragmatically Competent in Referring Expression Generation"
_colmweb.org/COLM/2025/Conference — COLM 2025_

### Official Review · Reviewer_tVi1 · 2025-05-13

**Rating:** 7
**Confidence:** 3
**Ethics Flag:** 1

**Summary:**

This work studies the Referring Expression Generation task in VLMs, aiming to understand the pragmatic competence of models in comparison with humans.
Using a dataset of 995 images with both human-written and human-spoken referring expressions, the authors identify that models often generate expressions that fail to uniquely identify the referent and may include excessive or irrelevant information.
Through a separate synthetic experiment, the authors also find that model-generated expressions differ from human ones in terms of which visual features are used.
Based on the overall evaluation results across 7 models and 4 types of metrics, this study also discusses the limitations of current automatic evaluation methods.

**Questions To Authors:**

1. While it might be nice-to-have, I think it could be interesting for the authors to share a bit more about the reasons behind the failure cases. Is it because the VLMs couldn’t capture the visual features well, or because they struggled with the language aspect of the task?

2. I also feel this would be a nice addition. Since the paper begins with motivation from the Gricean maxims, it would be interesting to see the distribution of model failures, either based on human validation or other metrics.

3. Line 226 typo: “listers” should be “listeners”.
4. In Table 1: The Human Spoken CIDEr score is listed as 115.64. Is it a typo?
5. Figures 6 and 7: It seems that p_hat is not defined.

**Reasons To Accept:**

The paper addresses an interesting and important question about VLMs' pragmatic competence through the REG task, which connects well with the Gricean maxims, i.e., for effective communication, the maxims of quantity, quality, relation, and manner matter more than mere accuracy.

The paper is easy to read, well-written, and clearly structured, with sufficient visual support.

**Reasons To Reject:**

Concerns about the experimental setup:

1. I didn’t quite understand lines 207–208 regarding the setup for the referring task. What is the justification for “randomly sampling one textual prompt variant”? For a controlled setting, wouldn’t it be better to use the same prompt for the entire dataset and run the experiment multiple times?
2. Following up on the previous point, the prompts seem very simple (in a single sentence), so I’m not sure whether the models fully understand the task.

Concerns about the results:
1. Besides the length distribution indicating that models may generate excessive or irrelevant expressions, are there any other quantitative results supporting this? For example, longer captions could be repetitive rather than truly irrelevant.
2. Regarding the human evaluation, is there any quality control in place? Was each model-generated expression evaluated by more than one annotator? While the task may not be very difficult, subjective factors could still influence human performance.

---

> ### Author Response · Authors · 2025-06-01
>
> ### W1-1: Prompting Diversity
> The purpose of randomly sampling one prompt variant per image-object pair is to introduce prompt diversity and reduce potential prompt-specific bias. While each test instance receives a different prompt, the setup remains controlled because we use the same prompt for a given image-object pair across all models, ensuring fair and consistent comparisons.
>
> ### W1-2: Prompting Length
> While the prompts are simple, each one clearly instructs the model to generate unambiguous expressions that uniquely identify the target object. Human subjects receive similar instructions to ensure fairness. Most models did recognize the task and produced relevant descriptions, but often failed at ensuring uniqueness, despite the prompt explicitly requiring it. We also found that longer prompts slightly degraded performance in smaller models, which motivated our choice of concise yet directive phrasing to avoid introducing unnecessary complexity.
>
> ### W2-1: Redundancy and Length
> Yes, we have further annotated the redundant parts in the expressions and found an average redundancy rate of 68.93% (based only on expressions evaluated as successful). Please see the results below.
>
> | Model | Instr. | Human | Redundancy % |
> |---|---|---|---|
> | LLaVA-7B | Dft. | 39.46 | 87.30 |
> |  | Brf. | 30.57 | 41.95 |
> | LLaVA-13B | Dft. | 46.40 | 91.85 |
> |  | Brf. | 32.53 | 49.65 |
> | LLaVA-34B | Dft. | 46.53 | 92.90 |
> |  | Brf. | 36.77 | 56.11 |
> | XComposer | Dft. | 52.19 | 92.81 |
> |  | Brf. | 51.65 | 80.36 |
> | MiniCPM-V | Dft. | 45.12 | 92.97 |
> |  | Brf. | 45.79 | 72.87 |
> | GLaMM | Dft. | 3.84 | 74.68 |
> |  | Brf. | 4.85 | 70.52 |
> | CogVLM | Dft. | 26.67 | 26.39 |
> |  | Brf. | 33.53 | 29.88 |
> | GPT-4o | Dft. | 59.80 | 89.81 |
> |  | Brf. | 51.72 | 52.75 |
> | Human | Spk. | 92.20 | 9.15 |
> |  | Wrt. | 89.29 | 7.29 |
>
>
> ### W2-2: Human Annotations
> As the reviewer noted, the task may not appear very difficult. However, we selected a subset of examples for annotation by 3 annotators and observed a Cohen’s kappa of κ = 0.781, indicating substantial agreement and validating the clarity and consistency of the task.
>
> ### Q1-2: Failure Cases
> We discussed failure cases in Section 5, including a representative example in Figure 8. We found that VLMs often fail to capture the most salient features, which are typically spatial expressions that humans rely on in language. Instead, the models tend to produce verbose combinations of visual attributes that are less preferred. On average, we observed a multiple-match rate of 35.82% and a no-match rate of 14.06%. Detailed results are provided below.
> | Model | Instr. | Wrong % | Multi. % | No-Mat % |
> |---|---|---|---|---|
> | LLaVA-7B | Dft. | 14.62 | 40.40 | 5.52 |
> |  | Brf. | 10.23 | 52.26 | 6.94 |
> | LLaVA-13B | Dft. | 26.26 | 26.20 | 1.14 |
> |  | Brf. | 10.30 | 56.63 | 0.54 |
> | LLaVA-34B | Dft. | 18.72 | 31.52 | 3.23 |
> |  | Brf. | 7.34 | 51.45 | 4.44 |
> | XComposer | Dft. | 20.20 | 24.92 | 2.69 |
> |  | Brf. | 14.28 | 31.45 | 2.62 |
> | MiniCPM-V | Dft. | 15.75 | 34.55 | 4.58 |
> |  | Brf. | 12.19 | 38.99 | 3.03 |
> | GLaMM | Dft. | 7.33 | 15.29 | 73.54 |
> |  | Brf. | 8.49 | 14.07 | 72.59 |
> | CogVLM | Dft. | 2.89 | 47.34 | 23.10 |
> |  | Brf. | 2.96 | 52.53 | 10.98 |
> | GPT-4o | Dft. | 11.98 | 24.04 | 4.18 |
> |  | Brf. | 10.97 | 31.52 | 5.79 |
> | Human | Spk. | 6.93 | 0.74 | 0.13 |
> |  | Wrt. | 7.68 | 2.36 | 0.67 |
>
>
> ### Q3-5: CIDEr Score and Typos
> This is not a typo. The score was computed using the official pycocoevalcap CIDEr implementation, with 3 human-written expressions as references per instance. Human-spoken expressions were highly consistent with the written ones, it is expected that the final score would exceed 100 in this case. We will revise other typos in the future version of the paper.

---

> > ### Comment · Reviewer_tVi1 · 2025-06-08
> >
> > Thanks for the clarification! Most of my concerns are addressed, so I increased the score. I'm not working closely with the REG tasks in the VLM context, so I’m keeping my confidence the same.

---

### Official Review · Reviewer_D8WT · 2025-05-13

**Rating:** 5
**Confidence:** 4
**Ethics Flag:** 1

**Summary:**

The authors study how modern VLMs perform on REG model. The authors collect a new dataset of 1k images and use that to evaluate how referring expressions generated by VLMs follow Grice maxims. The authors find that the VLMs are not good in terms of the pragmatics. The authors also show that the existing metrics are problematic for REG task.

**Reasons To Accept:**

- Pragmatics is largely neglected for referring expression generation task. It is good the authors try to raise attention on this.
- The authors propose curate the dataset carefully.
- The authors designed the evaluation carefully, including both different automatic evaluation as well as human evaluation.

**Reasons To Reject:**

- The authors use images from OpenImages dataset which could likely be in these VLM training data as well. Also, the authors only include images with multiple same category objects while it is still meaningful to include image with one such object which is also common in real world.
- The data collection tool used by the authors is still “region captioning” instead of the original ReferItGame which is more aligned with Grice maxim.
- The authors did not use chain of thought prompt nor use reasoning model to generate the referring expression. (Even at the time of submission, multi-modal reasoning model may not be available at the submission time, one can generate using some pipeline like: first generate captions for all objects in the image then use a reasoning model to reduce to one final expression)
- The conclusions in the paper are not surprising. The drawbacks of each type of evaluations have been discussed for a long time. And it is also not suprising that the models tend to provide lengthy captions because that is how the model as trained and encourage.

Reference:

ReferItGame: Referring to Objects in Photographs of Natural Scenes

---

> ### Author Response · Authors · 2025-06-01
>
> ### W1-1: Dataset Leakage
> We would like to clarify that our images are drawn from the validation split of the OpenImages segmentation task, as noted in Section 3.2. We trust VLM developers follow standard research practices and avoid using validation data for training or benchmark construction. In contrast, datasets like RefCOCO are built on the MSCOCO training set, which is known to suffer from data leakage. Our dataset design intentionally avoids this issue.
>
> ### W1-2: Single-Object Images
> Regarding the image selection criteria, we further include single-object images (492 images) in addition to the multi-object subset (995 images). These were evaluated in our extended experiments. Interestingly, we observed a sharp performance drop from an average accuracy of 69.45% on single-object images to 22.36% on images with multiple objects from the same category, highlighting the specific challenges posed by referential ambiguity rather than dataset composition alone.
>
> | Model | Instr. | Multi-Object | Single-Object | ΔAcc |
> |---|---|---|---|---|
> | LLaVA-7B | Dft. | 18.63 | 81.50 | −62.87 |
> |  | Brf. | 10.37 | 71.34 | −60.97 |
> | LLaVA-13B | Dft. | 28.80 | 81.91 | −53.11 |
> |  | Brf. | 10.67 | 76.63 | −65.96 |
> | LLaVA-34B | Dft. | 29.41 | 81.10 | −51.69 |
> |  | Brf. | 15.11 | 80.59 | −65.48 |
> | XComposer | Dft. | 37.56 | 81.70 | −44.14 |
> |  | Brf. | 35.55 | 84.15 | −48.60 |
> | MiniCPM-V | Dft. | 26.49 | 82.72 | −56.23 |
> |  | Brf. | 26.99 | 83.74 | −56.75 |
> | GLaMM | Dft. | 1.31 | 8.94 | −7.63 |
> |  | Brf. | 1.31 | 11.99 | −10.68 |
> | CogVLM | Dft. | 13.39 | 53.46 | −40.07 |
> |  | Brf. | 16.72 | 67.48 | −50.76 |
> | GPT-4o | Dft. | 48.14 | 83.33 | −35.19 |
> |  | Brf. | 37.36 | 80.69 | −43.33 |
> | Human | Spk. | 91.74 | 93.50 | −1.76 |
> |  | Wrt. | 88.31 | 91.26 | −2.95 |
>
>
> ### W2: Annotation
> Our pipeline is largely aligned with the ReferItGame setup, with the main difference being that we use red bounding boxes instead of red outlines to mark target objects (see Figure 9). Importantly, we focus on collecting unambiguous referring expressions that uniquely identify a target object in each image. This contrasts with region captioning, which permits general descriptions that may not even mention the object explicitly.
>
>
> ### W3: Chain of Thought
> As the reviewer noted, “multi-modal reasoning models may not be available at the submission time.” We emphasize our focus is to probe intuitive pragmatics in VLMs, as generating referring expressions is typically a fast, effortless process for humans. That said, we also include additional experiments where GPT-4o is explicitly prompted with CoT.
>
> | Model   | Instr.  | BLEU-1 | BLEU-4 | ROUGE-1 | ROUGE-L | METEOR | CIDEr  | SPICE | BERTScore | CLIPScore | REC   | Human |
> |---------|---------|--------|--------|---------|---------|--------|--------|--------|------------|-----------|--------|--------|
> | GPT-4o  | Dft. | 7.47   | 0.85   | 11.61   | 10.43   | 17.39  | 0.03   | 7.21   | 84.57      | 80.81     | 41.29 | 59.80 |
> |         | Brf. | 25.30  | 5.78   | 28.76   | 27.36   | 19.02  | 8.17   | 15.31  | 88.11      | 76.58     | 40.08 | 51.72 |
> |         | CoT     | 19.98  | 4.02   | 23.42   | 21.84   | 18.15  | 10.15  | 13.36  | 86.83      | 77.59     | 46.67 | 65.59 |
>
>
> ### W4: Not Surprising Results
> Our conclusions go well beyond the known limitations of existing evaluation metrics or the observation that VLMs tend to be verbose. As detailed in Section 5.1, we present human studies revealing a misalignment in pragmatic intuition between VLMs and humans. More importantly, our error analysis (see Table below) shows a 35.82% multiple-matches rate and a 14.06% no-match rate, demonstrating that models frequently fail to disambiguate between similar referents, which is a failure that reflects a lack of pragmatic competence, not just verbosity. The goal of our work is to recenter the REG task on its original pragmatic grounding, and to critically assess whether VLMs can handle this core linguistic challenge, which we argue has been overlooked in current evaluation practices.
>
> | Model | Instr. | Wrong % | Multi. % | No-Mat % |
> |---|---|---|---|---|
> | LLaVA-7B | Dft. | 14.62 | 40.40 | 5.52 |
> |  | Brf. | 10.23 | 52.26 | 6.94 |
> | LLaVA-13B | Dft. | 26.26 | 26.20 | 1.14 |
> |  | Brf. | 10.30 | 56.63 | 0.54 |
> | LLaVA-34B | Dft. | 18.72 | 31.52 | 3.23 |
> |  | Brf. | 7.34 | 51.45 | 4.44 |
> | XComposer | Dft. | 20.20 | 24.92 | 2.69 |
> |  | Brf. | 14.28 | 31.45 | 2.62 |
> | MiniCPM-V | Dft. | 15.75 | 34.55 | 4.58 |
> |  | Brf. | 12.19 | 38.99 | 3.03 |
> | GLaMM | Dft. | 7.33 | 15.29 | 73.54 |
> |  | Brf. | 8.49 | 14.07 | 72.59 |
> | CogVLM | Dft. | 2.89 | 47.34 | 23.10 |
> |  | Brf. | 2.96 | 52.53 | 10.98 |
> | GPT-4o | Dft. | 11.98 | 24.04 | 4.18 |
> |  | Brf. | 10.97 | 31.52 | 5.79 |
> | Human | Spk. | 6.93 | 0.74 | 0.13 |
> |  | Wrt. | 7.68 | 2.36 | 0.67 |

---

> > ### Comment · Reviewer_D8WT · 2025-06-10
> >
> > Thanks for the authors' reply.
> > W2: Annotation.
> > When I mean referitgame, i meant a two player cooperative process where the player score is based on speed and accuracy.
> > W3:
> > Generating referring expressions is typically a fast, effortless process for humans does not mean reasoning is not needed, especially thinking of cases when image is extremely crowd.
> >
> > I appreciate the authors to include new metrics, new data (single-image). I am raising my rating a bit but remain negative.

---

### Official Review · Reviewer_drYW · 2025-05-13

**Rating:** 6
**Confidence:** 3
**Ethics Flag:** 1

**Summary:**

The authors aim to address an issues that's often overlooked when evaluating VLMs: how practically useful the VLM outputs are to users. To this end, the authors construct a new evaluation dataset consisting of 995 images and human-written captions. Then they evaluated mainstream VLMs on a few aspects that may matter to human users.

**Questions To Authors:**

N/A

**Reasons To Accept:**

The problem is well motivated. Although achieving good performance using conventional evaluation metrics, VLMs tend to output verbose answers that reduce their usefulness. Moreover, when multiple objects of the same category appear in the image, VLM outputs may not focus on features that uniquely distinguish the target object.

**Reasons To Reject:**

1. The dataset size is quite small with only 995 images. This means the dataset can be easily overfit in the future.
2. Gemini 2.5 is not among the evaluated methods.
3. It's a bit overclaiming that current VLMs are not "pragmatically competent" in referring expression generation. The underwhelming performance is more like a misalignment between training and test. The performance degeneration due to misalignment can usually be fixed through finetuning for "pragmatic competence" and the finetuning probably won't take many iterations.

---

> ### Author Response · Authors · 2025-06-01
>
> We thank the reviewer for recognizing that the problem is well motivated. Below, we address the concerns:
>
> ### W1: Size and Overfit
> We would like to clarify that our dataset was originally intended for evaluation rather than training. Therefore, overfitting should not be a major concern, assuming standard research practices are followed and models are not trained on the test set.
>
>
> ### W2: Gemini 2.5
> At the time of our experiments, Gemini 2.5 was not available: its experimental release on March 25 came just days before our submission deadline on March 28. Additionally, the model declines to generate referring expressions when human faces are present in the image, which makes it hard to compare with other models.
>
> ### W3: Overclaiming
> While fine-tuning may mitigate some surface-level issues like verbosity, our results point to a deeper, more structural limitation. Specifically, we observed an average multiple-matches rate of 35.82% and a no-match rate of 14.06%, indicating that models often fail to disambiguate between similar referents, which is a core requirement of pragmatic competence. This failure is not easily addressed by a few iterations of fine-tuning, as it reflects a representational and inferential gap rather than a stylistic one. From the author’s prior experience pre-training VLMs, we found that addressing such failures typically requires more than just lightweight adjustments. Our aim is not to dismiss current VLMs, but to highlight the need to revisit the REG task itself, which has been underexplored despite its foundational role in evaluating referential language understanding.
>
> | Model | Instr. | Wrong % | Multi. % | No-Mat % |
> |---|---|---|---|---|
> | LLaVA-7B | Dft. | 14.62 | 40.40 | 5.52 |
> |  | Brf. | 10.23 | 52.26 | 6.94 |
> | LLaVA-13B | Dft. | 26.26 | 26.20 | 1.14 |
> |  | Brf. | 10.30 | 56.63 | 0.54 |
> | LLaVA-34B | Dft. | 18.72 | 31.52 | 3.23 |
> |  | Brf. | 7.34 | 51.45 | 4.44 |
> | XComposer | Dft. | 20.20 | 24.92 | 2.69 |
> |  | Brf. | 14.28 | 31.45 | 2.62 |
> | MiniCPM-V | Dft. | 15.75 | 34.55 | 4.58 |
> |  | Brf. | 12.19 | 38.99 | 3.03 |
> | GLaMM | Dft. | 7.33 | 15.29 | 73.54 |
> |  | Brf. | 8.49 | 14.07 | 72.59 |
> | CogVLM | Dft. | 2.89 | 47.34 | 23.10 |
> |  | Brf. | 2.96 | 52.53 | 10.98 |
> | GPT-4o | Dft. | 11.98 | 24.04 | 4.18 |
> |  | Brf. | 10.97 | 31.52 | 5.79 |
> | Human | Spk. | 6.93 | 0.74 | 0.13 |
> |  | Wrt. | 7.68 | 2.36 | 0.67 |

---

> > ### Comment · Reviewer_drYW · 2025-06-04
> > **multimatch**
> >
> > I've tried an example image (Figure 8 in the paper) with the multimatch issue. Just ask gpt-4o with this prompt: "describe the object in the blue bbox with words that can discriminate it with similar objects", it will output a few distinctive features including: 1. Cracked surface with prominent sugar coating, 2. Dark cocoa-brown coloration, 3. Positioned front and center, 4. High textural detail (sharp cracks, uneven surface). Although the output is a bit verbose, at least point 3 is consistent with human annotations. I still tend to think that the alignment issue is easy to fix, either with special prompts or with careful finetunings. Therefore, I'd like to maintain my rating.

---

### Official Review · Reviewer_GKs2 · 2025-05-22

**Rating:** 7
**Confidence:** 3
**Ethics Flag:** 2

**Summary:**

The paper investigates the effectiveness of vision-language models (VLMs) for the task of referring expression generation (REG); specifically whether expressions generated by VLMs adhere to the cooperative principles (i.e. Gricean maxims) and align with human-annotated expressions. To that end, the authors curate a new dataset spanning (i) both COCO and non-COCO classes to address data leakage and (ii) synthetic instances to illustrate whether humans and VLMs attend to different pragmatic cues. To ensure pragmatic competency the dataset is annotated with both spoken and written referring expressions. The dataset is used to systematically evaluate the REG capabilities of a wide range of VLMs and identify three crucial pragmatic failures, namely (i) the inability to uniquely identify objects, (ii) the inability to refer with minimal required information and (iii) the inability to align with human pragmatic preferences. Their comprehensive evaluation suite also sheds light on the pitfalls of automated evaluation metrics for this task of REG.

**Questions To Authors:**

See above

**Reasons To Accept:**

1. The paper is well written and relevant to the current discourse of understanding how LLMs and broadly AI systems differ in pragmatic competence with respect to humans.

2. The paper provides a comprehensive literature review of the task and helps situate this work in the broader context of pragmatics, capabilities of visual language models, referring to expression comprehension amongst others. A minor detail that the reviewer applauds is how each statement or proposition put forward is accompanied by a relevant citation such as restoring the pragmatic focus of the REG task as its original formulation, the difference between written and spoken language in human communication.

3. The experimental suite is comprehensive spanning different categories of VLMs such as open-source and proprietary, SFT vs SFT + RLHF, and mechanistically grounded VLMs amongst others. They also explore different prompting types and a wide host of evaluation metrics. The experimental design scores high on soundness.

4. They create a new dataset to address concerns of dataset contamination and also synthetic instances to investigate how VLMs differ in pragmatic competence from humans.

5. Their findings and analyses are insightful and shed light on the different pragmatic failures of VLMs as well as the limitations of current automated evaluation metrics for this task of REG.

**Reasons To Reject:**

1. The paper highlights that VLMs by themselves are unable to adhere to the cooperative principles of human communication, using the simple zero-shot prompting strategy (brief or detailed).  It would be interesting to observe whether the generations would change if (i) the models were explicitly instructed to adhere to Gricean Maxims and/ or all the co-operative principles , (ii) the models were able to refine the output by verifying whether the generated expression flouts any of the given maxims, or (iii) simply provided with a few few-shot cases to calibrate the generations during inference.

2. The paper plays a great emphasis on creating a new dataset comprising both COCO and non-COCO classes to address concerns of dataset contamination as in other REG datasets. However, the results are presented at an aggregate level and not over the old/new classes. Do models far better on objects corresponding to COCO-classes or do they demonstrate similar performance?

3. Likewise, while the authors employ a human-centered evaluation study to assess the pragmatic limitations of automated REGs, the paper does not provide details of the error cases. What fraction of cases have redundant information and/or is not specific enough? Based on information in Figure 10, if the annotator chooses the option that they cannot tell where the object is or there are multiple matches, does the score automatically become 0. Furthermore, what is the distribution of these decision choices across different VLMs?

4. The paper relies on the Appendix to clarify additional information pertaining to the breadth of the visual saliencies explored to better explain Figures 5, 6, and 7. Also there are a few typographic errors in the heatmaps (Figures 11-14).

---

> ### Author Response · Authors · 2025-06-01
>
> We thank the reviewer for recognizing our thorough and well-cited analysis of pragmatic competence in VLMs through a carefully designed experimental suite and novel dataset, yielding insightful findings on both model limitations and evaluation metrics. Below, we address the concerns:
>
> Note that the results presented in our original submission were based solely on multi-object images (approximately 1,000). In subsequent experiments, we included an additional set of single-object images (approximately 500), and all updated results now reflect performance on both single- and multi-object images.
>
> ### W1: Additional Prompting
>
> We emphasize our focus on intuitive pragmatics, as generating referring expressions is typically a fast, effortless process for humans. That said, we also include additional experiments where GPT-4o is explicitly prompted to follow Gricean principles.
> | Model   | Instr.   | BLEU-1 | BLEU-4 | ROUGE-1 | ROUGE-L | METEOR | CIDEr  | SPICE | BERTScore | CLIPScore | REC   | Human |
> |---------|----------|--------|--------|---------|---------|--------|--------|--------|------------|-----------|--------|--------|
> | GPT-4o  | Dft.  | 7.47   | 0.85   | 11.61   | 10.43   | 17.39  | 0.03   | 7.21   | 84.57      | 80.81     | 41.29 | 59.80 |
> |         | Brf.  | 25.30  | 5.78   | 28.76   | 27.36   | 19.02  | 8.17   | 15.31  | 88.11      | 76.58     | 40.08 | 51.72 |
> |         | Gricean  | 21.43  | 4.33   | 25.62   | 24.01   | 20.35  | 10.85  | 13.87  | 87.57      | 78.50     | 42.10 | 63.50 |
>
>
> ### W2: Performance Gap in Non-COCO Classes
> We have conducted such further analysis and observed performance differences across nearly all models between COCO classes and non-COCO classes. Please see the detailed results below:
> | Model | Instr. | COCO | No-COCO | ΔAcc |
> |---|---|---|---|---|
> | LLaVA-7B | Dft. | 41.26 | 37.65 | −3.61 |
> |  | Brf. | 31.18 | 29.96 | −1.22 |
> | LLaVA-13B | Dft. | 45.70 | 47.10 | 1.40 |
> |  | Brf. | 33.47 | 31.58 | −1.89 |
> | LLaVA-34B | Dft. | 48.25 | 44.80 | −3.45 |
> |  | Brf. | 38.04 | 35.49 | −2.55 |
> | XComposer | Dft. | 56.05 | 48.31 | −7.74 |
> |  | Brf. | 55.78 | 47.50 | −8.28 |
> | MiniCPM-V | Dft. | 47.98 | 42.24 | −5.74 |
> |  | Brf. | 49.46 | 42.11 | −7.35 |
> | GLaMM | Dft. | 4.30 | 3.37 | −0.93 |
> |  | Brf. | 4.30 | 5.40 | 1.10 |
> | CogVLM | Dft. | 27.96 | 25.37 | −2.59 |
> |  | Brf. | 34.81 | 32.25 | −2.56 |
> | GPT-4o | Dft. | 63.31 | 56.28 | −7.03 |
> |  | Brf. | 54.84 | 48.58 | −6.26 |
> | Human | Spk. | 92.07 | 92.58 | 0.51 |
> |  | Wrt. | 89.52 | 89.07 | −0.45 |
>
>
> ### W3: Error Analysis
> If the annotator chooses the option that they cannot tell where the object is or there are multiple matches, the score will be 0.
> We observed an average multiple-matches rate of 35.82% and a no-match rate of 14.06%. We also manually annotated the redundant information in the referring expressions and found a redundancy rate of 68.93%. Detailed results are listed below.
> | Model | Instr. | Wrong % | Multi. % | No-Mat % |
> |---|---|---|---|---|
> | LLaVA-7B | Dft. | 14.62 | 40.40 | 5.52 |
> |  | Brf. | 10.23 | 52.26 | 6.94 |
> | LLaVA-13B | Dft. | 26.26 | 26.20 | 1.14 |
> |  | Brf. | 10.30 | 56.63 | 0.54 |
> | LLaVA-34B | Dft. | 18.72 | 31.52 | 3.23 |
> |  | Brf. | 7.34 | 51.45 | 4.44 |
> | XComposer | Dft. | 20.20 | 24.92 | 2.69 |
> |  | Brf. | 14.28 | 31.45 | 2.62 |
> | MiniCPM-V | Dft. | 15.75 | 34.55 | 4.58 |
> |  | Brf. | 12.19 | 38.99 | 3.03 |
> | GLaMM | Dft. | 7.33 | 15.29 | 73.54 |
> |  | Brf. | 8.49 | 14.07 | 72.59 |
> | CogVLM | Dft. | 2.89 | 47.34 | 23.10 |
> |  | Brf. | 2.96 | 52.53 | 10.98 |
> | GPT-4o | Dft. | 11.98 | 24.04 | 4.18 |
> |  | Brf. | 10.97 | 31.52 | 5.79 |
> | Human | Spk. | 6.93 | 0.74 | 0.13 |
> |  | Wrt. | 7.68 | 2.36 | 0.67 |
>
>
> ### W4: Presentations
> We apologize for the oversight. We will correct these errors in our final version.

---

> > ### Comment · Reviewer_GKs2 · 2025-06-02
> > **Response to Authors**
> >
> > Thank you Authors for responding to my comments. I appreciate the timely response and the additional analysis. I hope you can incorporate them into the final camera-ready version.
> >
> > Could you clarify how did you compute this redundancy rate? Is it based on the proportion of tokens in the REs or the fraction of REs which have some redundant information.
> >
> > Likewise, could you also clarify the difference between Wrong, No Match, or Multiple Match. How is it related to the main Results in Table 1? Is Wrong % is the fraction of cases human evaluation got the answer incorrect, either because (i) they clicked that they could not find the answer, (ii) they clicked there were multiple answers or (iii) they clicked on the wrong object. Or is wrong % only (iii).   If it is the latter, it is surprising that the summation of these three percentages and the proportion of correct answers (such as  18.59 for LLava-7B in Table 1) does not add up to 100%.
> >
> > I hope you could clarify this further.

---

> > > ### Author Response · Authors · 2025-06-02
> > >
> > > > Thank you Authors for responding to my comments. I appreciate the timely response and the additional analysis. I hope you can incorporate them into the final camera-ready version.
> > >
> > > Thank you! We will make sure to incorporate the revisions and additional analyses into the final camera-ready version.
> > >
> > > > Could you clarify how did you compute this redundancy rate? Is it based on the proportion of tokens in the REs or the fraction of REs which have some redundant information.
> > >
> > > We annotate the minimal set of text spans necessary to uniquely identify the referent. Any remaining words in the referring expression are considered redundant. The redundancy rate is then calculated as the proportion of these redundant words relative to the total number of words in each referring expression.
> > >
> > > > Likewise, could you also clarify the difference between Wrong, No Match, or Multiple Match. How is it related to the main Results in Table 1? Is Wrong % is the fraction of cases human evaluation got the answer incorrect, either because (i) they clicked that they could not find the answer, (ii) they clicked there were multiple answers or (iii) they clicked on the wrong object. Or is wrong % only (iii).
> > >
> > > You're correct that “Wrong” refers specifically to case (iii), where the annotator clicked on the wrong object.
> > >
> > > > If it is the latter, it is surprising that the summation of these three percentages and the proportion of correct answers (such as 18.59 for LLava-7B in Table 1) does not add up to 100%.
> > >
> > > Thank you for catching the discrepancy in the percentages.
> > > This is due to a revision in our dataset: in response to Reviewer D8WT’s suggestion, we added 492 images containing objects with themselves as the only instance from their classes (and we also removed 2 images due to safety and fairness concerns, which does not change the results much). The statistics we report in this response are based on this updated dataset, rather than the original one, which only included multi-instance scenes.
> > > For example, after the dataset augmentation, the percentage of correct identifications becomes 39.46% (18.63% for multi-instance, 81.50% for single-instance). Along with 14.62% (Wrong), 40.40% (Multiple Match), and 5.52% (No Match), these add up to 100%.
> > > **To present the updated results more clearly, we’ve included two summary tables in our general response.**
> > >
> > > ---
> > > We hope these fully address your concerns, and please let us know if any further information can clarify!

---

> > > > ### Comment · Reviewer_GKs2 · 2025-06-04
> > > > **Response to Updated Results**
> > > >
> > > > Thank you for the updated results and for providing the detailed error breakdown in the General Response comment.
> > > >
> > > > I will maintain my score but will also be willing to defend it since I feel the work has merits and is experimentally sound and rigorous in its findings.

---

### Author Response · Authors · 2025-06-02
**General Response**

## General Update on Dataset and Main Results
We thank all reviewers for their insightful feedback and suggestions. We have conducted further experiments and expanded our evaluation dataset (thanks to Reviewer D8WT’s suggestion) to include 492 single-object images in addition to the original 993 multi-object images (removed 2 of them due to safety/fairness concerns). This allows for a more comprehensive assessment of model performance across varied pragmatic settings.

Accordingly, we have updated the main results table, which now reflects performance across the full set of 1,485 images. Irrel% stands for Redundant/Irrelevant Words%. We annotate the minimal set of text spans necessary to uniquely identify the referent. Any remaining words in the referring expression are considered redundant. The redundancy rate is then calculated as the proportion of these redundant words relative to the total number of words in each referring expression.

| Model       | Instr. | BLEU-1 | BLEU-4 | ROUGE-1 | ROUGE-L | METEOR | CIDEr | SPICE | BERT   | CLIP   | REC   | Human | Irrel% |
|-------------|--------|--------|--------|---------|---------|--------|--------|--------|--------|--------|--------|--------|--------|
| LLaVA-7B    | Dft.   | 13.27  | 1.60   | 18.09   | 16.30   | 19.29  | 2.10   | 10.50  | 85.51  | 79.02  | 32.41 | 39.46 | 87.30  |
|             | Brf.   | 28.74  | 6.05   | 36.46   | 35.50   | 19.15  | 10.80  | 24.59  | 89.02  | 70.72  | 25.51 | 30.57 | 41.95  |
| LLaVA-13B   | Dft.   | 8.17   | 1.07   | 11.98   | 10.94   | 16.89  | 0.77   | 7.92   | 84.61  | 79.85  | 30.13 | 46.40 | 91.85  |
|             | Brf.   | 28.96  | 5.81   | 36.44   | 35.64   | 20.13  | 8.14   | 21.63  | 88.42  | 72.99  | 28.92 | 32.53 | 49.65  |
| LLaVA-34B   | Dft.   | 6.29   | 0.78   | 9.82    | 9.11    | 16.15  | 0.07   | 7.61   | 84.39  | 79.86  | 33.42 | 46.53 | 92.90  |
|             | Brf.   | 28.55  | 6.38   | 32.99   | 31.67   | 20.48  | 9.60   | 16.50  | 88.50  | 74.95  | 35.24 | 36.77 | 56.11  |
| XComposer   | Dft.   | 5.25   | 0.65   | 8.38    | 7.81    | 14.58  | 3.10   | 6.37   | 84.11  | 79.86  | 38.06 | 52.19 | 92.81  |
|             | Brf.   | 13.59  | 2.17   | 17.77   | 16.69   | 19.95  | 5.52   | 10.63  | 85.52  | 79.66  | 38.47 | 51.65 | 80.36  |
| MiniCPM-V   | Dft.   | 6.38   | 0.67   | 9.86    | 8.78    | 15.28  | 0.05   | 6.30   | 84.29  | 80.38  | 37.93 | 45.12 | 92.97  |
|             | Brf.   | 16.03  | 3.15   | 19.56   | 18.19   | 18.77  | 6.36   | 11.16  | 86.29  | 78.55  | 35.04 | 45.79 | 72.87  |
| GLaMM       | Dft.   | 15.01  | 3.32   | 16.69   | 16.29   | 11.49  | 9.08   | 3.90   | 86.42  | 58.26  | 5.78  | 3.84  | 74.68  |
|             | Brf.   | 18.46  | 4.45   | 20.92   | 20.46   | 14.18  | 10.48  | 4.44   | 86.65  | 58.60  | 5.72  | 4.85  | 70.52  |
| CogVLM      | Dft.   | 31.13  | 8.70   | 33.89   | 32.32   | 23.50  | 41.62  | 24.09  | 89.78  | 66.54  | 33.29 | 26.67 | 26.39  |
|             | Brf.   | 31.39  | 8.69   | 34.70   | 32.94   | 24.87  | 41.41  | 24.74  | 90.00  | 69.15  | 38.80 | 33.53 | 29.88  |
| GPT-4o      | Dft.   | 7.47   | 0.85   | 11.61   | 10.43   | 17.39  | 0.03   | 7.21   | 84.57  | 80.81  | 41.29 | 59.80 | 89.81  |
|             | Brf.   | 25.30  | 5.78   | 28.76   | 27.36   | 19.02  | 8.17   | 15.31  | 88.11  | 76.58  | 40.08 | 51.72 | 52.75  |
| Human       | Spk.   | 66.18  | 22.58  | 70.15   | 66.45   | 48.28  | 112.04 | 42.35  | 93.89  | 71.60  | 64.56 | 92.20 | 9.15   |
|             | Wrt.   | -      | -      | -       | -       | -      | -      | -      | -      | 70.43  | 63.69 | 89.29 | 7.29   |

(to be continued)

---

> ### Author Response · Authors · 2025-06-02
>
> ## Breakdown Analysis of Main Results
> We further provide a detailed breakdown of the main results.
>
> **Column Explanations:**
> - **REC**: The accuracy of the REC model (CogVLM-Grounding). We use a listener-based evaluation where referential success is defined by whether the model's predicted bounding box has an IoU > 0.5 with the ground truth.
> - **Human**: The accuracy of human listeners.
> - **Agree**: The agreement between the two listeners—the REC model and humans.
> - **Wrong%**: The proportion of failed human guesses, where the clicked location falls outside the target object’s segmentation mask.
> - **Multi.%**: The proportion of cases where the human listener indicated that the expression was ambiguous, i.e., it matched multiple objects in the image.
> - **No-Mat%**: The proportion of cases where the human listener reported that no object matched the expression.
> - **COCO / No-COCO**: Accuracy on objects that are part of COCO classes and those that are outside of it, respectively.
> - **ΔAcc (Class)**: The drop in accuracy between COCO and non-COCO objects.
> - **Coocc. / No-Coocc.**: Accuracy on images containing multiple objects of the same class (co-occurring) and those containing only one instance of the target class.
> - **ΔAcc (Co-occurrence)**: The drop in accuracy between co-occurring and non-co-occurring cases.
>
> | Model | Instr. | Listener&nbsp;Compare |  |  | Error&nbsp;Breakdown |  |  | Class&nbsp;Breakdown |  |  | Class&nbsp;Co&#8209;occurrence |  |  |
> |---|---|---|---|---|---|---|---|---|---|---|---|---|---|
> |       |        | **Human** | **REC** | **Agree** | **Wrong %** | **Multi. %** | **No-Mat %** | **COCO** | **No-COCO** | **ΔAcc** | **Coocc.** | **No-Coocc.** | **ΔAcc** |
> | LLaVA-7B | Dft. | 39.46 | 32.41 | 65.84 | 14.62 | 40.40 | 5.52 | 41.26 | 37.65 | -3.61 | 18.63 | 81.50 | -62.87 |
> |  | Brf. | 30.57 | 25.51 | 71.62 | 10.23 | 52.26 | 6.94 | 31.18 | 29.96 | -1.22 | 10.37 | 71.34 | -60.97 |
> | LLaVA-13B | Dft. | 46.40 | 30.13 | 65.10 | 26.26 | 26.20 | 1.14 | 45.70 | 47.10 | 1.40 | 28.80 | 81.91 | -53.11 |
> |  | Brf. | 32.53 | 28.92 | 67.99 | 10.30 | 56.63 | 0.54 | 33.47 | 31.58 | -1.89 | 10.67 | 76.63 | -65.96 |
> | LLaVA-34B | Dft. | 46.53 | 33.42 | 62.14 | 18.72 | 31.52 | 3.23 | 48.25 | 44.80 | -3.45 | 29.41 | 81.10 | -51.69 |
> |  | Brf. | 36.77 | 35.24 | 65.03 | 7.34 | 51.45 | 4.44 | 38.04 | 35.49 | -2.55 | 15.11 | 80.59 | -65.48 |
> | XComposer | Dft. | 52.19 | 38.06 | 66.11 | 20.20 | 24.92 | 2.69 | 56.05 | 48.31 | -7.74 | 37.56 | 81.70 | -44.14 |
> |  | Brf. | 51.65 | 38.47 | 64.09 | 14.28 | 31.45 | 2.62 | 55.78 | 47.50 | -8.28 | 35.55 | 84.15 | -48.60 |
> | MiniCPM-V | Dft. | 45.12 | 37.93 | 66.38 | 15.75 | 34.55 | 4.58 | 47.98 | 42.24 | -5.74 | 26.49 | 82.72 | -56.23 |
> |  | Brf. | 45.79 | 35.04 | 63.62 | 12.19 | 38.99 | 3.03 | 49.46 | 42.11 | -7.35 | 26.99 | 83.74 | -56.75 |
> | GLaMM | Dft. | 3.84 | 5.78 | 93.61 | 7.33 | 15.29 | 73.54 | 4.30 | 3.37 | -0.93 | 1.31 | 8.94 | -7.63 |
> |  | Brf. | 4.85 | 5.72 | 93.34 | 8.49 | 14.07 | 72.59 | 4.30 | 5.40 | 1.10 | 1.31 | 11.99 | -10.68 |
> | CogVLM | Dft. | 26.67 | 33.29 | 73.30 | 2.89 | 47.34 | 23.10 | 27.96 | 25.37 | -2.59 | 13.39 | 53.46 | -40.07 |
> |  | Brf. | 33.53 | 38.80 | 68.53 | 2.96 | 52.53 | 10.98 | 34.81 | 32.25 | -2.56 | 16.72 | 67.48 | -50.76 |
> | GPT-4o | Dft. | 59.80 | 41.29 | 62.00 | 11.98 | 24.04 | 4.18 | 63.31 | 56.28 | -7.03 | 48.14 | 83.33 | -35.19 |
> |  | Brf. | 51.72 | 40.08 | 63.01 | 10.97 | 31.52 | 5.79 | 54.84 | 48.58 | -6.26 | 37.36 | 80.69 | -43.33 |
> | Human | Spk. | 92.20 | 64.56 | 64.96 | 6.93 | 0.74 | 0.13 | 92.07 | 92.58 | 0.51 | 91.74 | 93.50 | -1.76 |
> |  | Wrt. | 89.29 | 63.69 | 63.69 | 7.68 | 2.36 | 0.67 | 89.52 | 89.07 | -0.45 | 88.31 | 91.26 | -2.95 |

---

### Decision · Program_Chairs · 2025-07-08

**Decision:**

Accept

**Comment:**

This paper explores the failures of VLMs in producing pragmatically-competent references through a systematic evaluation and analysis of key failures of existing models. Contributions cover a comprehensive evaluation suite (and experimental results following that suite) and a new dataset for evaluating this task. Additionally, the authors conducted existing prompting-based experiments in the rebuttals that tested out different prompts the reviewers suggested would be successful at the task (and weren't). The paper is well-written and timely.